# Performance Evaluation of Precast Concrete Using Microwave Heating Form

**DOI:** 10.3390/ma12071113

**Published:** 2019-04-03

**Authors:** Hyeonggil Choi, Taehoon Koh, Heesup Choi, Yukio Hama

**Affiliations:** 1School of Architecture, Kyungpook National University, Daegu 41566, Korea; hgchoi@knu.ac.kr; 2Advanced Railroad Civil Engineering Division, Korea Railroad Research Institute, Uiwang 16105, Korea; 3Department of Civil Engineering, Kitami Institute of Technology, Kitami, Hokkaido 090-8507, Japan; hs-choi@mail.kitami-it.ac.jp; 4College of Environmental Technology, Graduate School of Engineering, Muroran Institute of Technology, Muroran, Hokkaido 080-8585, Japan; hama@mmm.muroran-it.ac.jp

**Keywords:** microwave heating system, precast concrete, steam curing, site precast, demolding strength

## Abstract

The purpose of this study is to evaluate the temperature distribution, strength development, porosity, scanning electron microscopy observation, shrinkage, and surface properties of concrete in order to apply microwave heat curing to the precast method and to analyze the CO_2_ emissions and economic feasibility of microwave heat curing. The heating of a steel form by microwave heating enabled concrete to be efficiently cured at a temperature within a range of ±5 °C. After the curing, demolding strength could be cleared through the densification of the concrete by decreasing the porosity of the concrete. Microwave heat curing exhibited excellent performance compared to conventional steam curing in terms of efficient temperature control, occurrence of cracks due to shrinkage, surface condition of concrete after curing, economic efficiency, and CO_2_ emissions. However, verification and supplementation based on actual data are necessary so that environments applicable to the various sizes and shapes of forms can be prepared.

## 1. Introduction

In building construction, the amount of air and energy used should be saved while maintaining stable quality. The precast concrete (PCa) method, which is used to manufacture concrete products in factories and construct them in the field, is widely used in concrete construction. The PCa method provides various advantages such as quality, safety, and a reduction in construction time and labor. However, it poses the problem of environmental load and high cost owing to transportation [1,2]. To solve these problems, the site PCa method, which is the process of manufacturing PCa at a site and immediately constructing it, has attracted attention in recent years [3,4]. The site PCa method is shown in Figure 1. The disadvantages of this method are that a large space for a factory to manufacture the PCa products in the field, a curing facility, and a yard of produced PCa products are required. However, there is no requirement to transport large PCa products and it is possible to efficiently overcome the limitations on the order quantity of PCa plants. Therefore, it is highly likely that the PCa method will be replaced by the site PCa method in the future.

Currently, most PCa products are produced by accelerated curing using steam curing. However, steam curing requires a steam generator for curing, exhibits low thermal efficiency, and experiences problems, such as the deterioration of steel forms and poor working environments owing to the steam [5,6]. 

This study developed a new method of efficiently curing concrete using microwave (MW) heat in the minimum space. The MW heating technology provides advantages, such as rapid and uniform heating and high thermal efficiency, which can be utilized in various industrial fields [7,8,9]. Figure 2 shows the heating of a steel form using the MW heating technology. An MW absorber is stimulated to generate heat using 2.45 GHz MWs, and this heat is transferred to the steel form. This is a method of promoting the curing of concrete. Compared to the conventional steam curing used in the PCa system, the space utilization of this method is excellent and energy efficiency is significantly improved. Therefore, the application of the site PCa method can be expected to increase.

The purpose of this study was to compare and evaluate the temperature distribution, strength development, porosity, scanning electron microscopy (SEM) observation, shrinkage, surface condition, CO_2_ emissions, and the economic efficiency of concrete between conventional steam curing and MW heat curing. 

## 2. Experimental Method 

### 2.1. Outline of Experiment

The experimental conditions, materials, and the chemical composition of the cement and concrete mix are shown in Table 1, Table 2, Table 3 and Table 4. Experiments were carried out in four conditions that comprised one steam curing condition, which is applied in the PCa method, and three MW heat curing conditions. The conditions were different in terms of precuring time, heating time, cooling time, and maximum temperatures, as shown in Figure 3.

The concrete used in the experiment was a concrete mix with a water-to-cement ratio (W/C) of 42.6%, a target slump of 18.0 ± 2.5 mm, and a target air volume of 4.5 ± 1.5%. Ordinary Portland cement was used. Fine aggregate and coarse aggregate (maximum dimension: 20 mm) were used for crushed sand and limestone crushed stone, and chemical admixtures were used for the AE (air entraining) water reducing agent and AE agent.

### 2.2. Experimental Method

#### 2.2.1. Evaluation of the Basic Properties of Applied Concrete

To evaluate the basic properties of the applied concrete, we considered fresh concrete properties and carried out strength development and free shrinkage tests. The MW3 specimen was employed for evaluating the strength and shrinkage characteristics because the same concrete was used in the four tests. After concrete pouring, seal curing was performed at 20 °C and 60% relative humidity, and then the form was removed on the first day. Compressive strength was measured at 1, 7, 14, and 28 days of age at 20 °C under water curing using a circumferential test specimen of φ100 mm × 200 mm according to JIS A 1108 [10]. The free shrinkage test was carried out according to JIS A 1129-3 [11] until 28 days. The specimen used was a 100 mm × 100 mm × 400 mm prism, which was placed in a constant temperature and humidity chamber at 20 °C and 60% RH after 1 week of curing in water at 20 °C [11].

#### 2.2.2. Performance Evaluation of Steam Curing and MW Heat Curing in a Mock-Up Sample

Mock-up tests were carried out to evaluate the performance of concrete with steam curing and MW heat curing. Figure 4 shows the simulated specimen, and Figure 5 shows the experimental setup. A steel form with a width of 6000 mm, a length of 500 mm, and a height of 200 mm was manufactured and tested.

To generate heat uniformly in MW heat curing, four MW generators were installed in the lower part of the steel form through preliminary examination so that heat was uniformly transmitted from the lower part of the steel form. In the mock-up test, the test specimens with a total length of 6000 mm were divided into two parts and tested with reinforced concrete (3000 mm in width) and plain concrete (3000 mm in width). In addition, for core sampling, the normal concrete was divided into two parts with a width of 1500 mm and experiments were conducted for each part.

The temperature distribution in the concrete was measured immediately after concrete pouring by a data logger after installing a thermocouple at three locations on the upper, middle, and lower side of the surface on which the concrete was poured. The surface temperature of the concrete was measured by a thermography camera immediately after removing the form.

Columnar core specimens of φ100 mm × 200 mm were collected in a balanced manner immediately after removing the form and at 1, 7, and 14 days of age, as shown in Figure 4. The compressive strength and porosity of the specimens were determined by a mercury intrusion porosimeter (MIP). Hydration products were observed through SEM using a specimen with an age of 14 days. The specimens for porosity measurement and SEM observation were sampled from the core specimen, and the test samples were cured in water for 4 weeks at 20 °C. For the MIP test, shown in Figure 6, in order to obtain samples having possible representativeness for measurement of porosity, samples were collected for each age and cut into cubes of 5 mm and immersed in ethanol for 1 week to stop the hydration of cementitious material. After that, drying by the D-dry method was carried out for 1 week, and then the pore distribution of each MIP sample was measured by the mercury intrusion porosimeter (MIP; Quantachrome PoreMaster 33, anton paar Ltd., Tokyo, Japan) to a minimum diameter of 6 nm with a maximum pressure of 220 MPa. Also, for the SEM test, shown in Figure 7, samples were collected for age 14 days and, similar to the MIP method, immersed in ethanol for 1 week to stop the hydration of cementitious material. After that, drying by the D-dry method was carried out for 1 week, and then the hydration products of each SEM sample with platinum coating was measured using a scanning electron microscope (SEM, JSM-6380, Jeol Ltd., Tokyo, Japan) with an accelerating voltage of 15 kV.

Free shrinkage and restraint shrinkage were measured by a data logger, starting from immediately after concrete pouring until 28 days of age, using reinforcing bars (D10, 600 mm in settlement length) with strain gauges attached on both sides of the concrete and reinforced concrete. In addition, the surface permeability coefficient of concrete was measured using the Torrent method, and the state of the concrete surface was examined at an age of 28 days.

CO_2_ emissions and curing energy costs were calculated according to the guidelines provided by the Japanese Ministry of the Environment, heavy fuel oil charges, and the amount of electric power and electricity used by the business [12,13].

## 3. Experimental Results and Discussion

### 3.1. Basic Properties of Applied Concrete

Table 5 shows the fresh concrete properties and compressive strength of the applied concrete. It can be seen that the target slump and the target air content were achieved in the four tests. The same concrete mix was used in each experiment. Based on the results of the slump and air content, the applied concrete was considered to have the same performance in the four tests.

The compressive strength exceeded the design strength of 30 MPa at 7 days of age. In addition, it exceeded 10 MPa at 28 days of age compared to the design strength of 30 MPa.

The free shrinkage and mass loss ratio are shown in Figure 8. As the rate of change in mass increased, the amount of shrinkage increased and reached 333 μ at 28 days of age. The concrete applied in this experiment corresponded to the concrete specification of concrete grade classification (less than 500 μ) according to drying shrinkage deformation [14]. This implies that the concrete applied in this experiment corresponded to a concrete with low W/C and with limestone aggregate, which is effective for shrinkage reduction.

### 3.2. Performance of Concrete Mock-Up Samples under Steam Curing and MW Heat Curing

#### 3.2.1. Temperature Distribution

Figure 9 and Figure 10 show the internal temperature distribution of concrete during steam curing and MW heat curing (MW2). In the case of steam curing, there was no difference between the temperatures at various measurement intervals. However, the highest temperature was 30 °C higher than the set temperature (60 °C). In addition, in MW heat curing, heat was transferred from the lower part of the steel form. Thus, the temperature difference between the upper and lower parts was approximately 9 °C. However, the temperature at the center could be controlled within a range of ± 5 °C.

The temperature distribution on the concrete surface obtained by a thermography camera immediately after removing the form is shown in Figure 11. There was a small difference of approximately 4 °C in the surface temperature between steam curing and MW heat curing. Further, it was confirmed that the heat was uniformly generated.

#### 3.2.2. Strength Properties (Core Strength)

Figure 12 shows the compressive strength of the core specimen, which was a simulated member. Steam curing exhibited higher initial strength compared to MW heat curing. However, it was less than 10% after 14 days because of the small increase in strength after steam curing. Figure 13 shows the relationship between the total temperature and compressive strength after removing the form. Concrete strength was strongly influenced by the curing temperature. In the case of steam curing, the initial strength increased because the initial curing temperature was high. However, the initial curing temperature should not be increased sharply because it decreases long-term durability and causes problems, such as the occurrence of cracks and a decrease in strength development. In addition, the demolding strength of spotted concrete (5 MPa) specified in JASS5 of the Architectural Institute of Japan is exceeded in steam curing and MW heat curing at the time of removing the form. However, a demolding strength of 10 MPa or more is required considering the onsite environment of large-sized PCa product movement at PCa factories, etc. In addition, it was considered that the curing condition of MW3 is more advantageous based on the strength development after demolding and the durability degradation in the experimental conditions.

#### 3.2.3. Porosity and SEM Observation

Figure 14 and Figure 15 show the measurement results of pore and cumulative pore distribution, respectively, according to the age of the steam curing and MW heat curing by mercury intrusion porosimeter (MIP). Also, three MIP samples were collected per specimen of each case, and a total of three repeat measurements were performed. As a result, three MIP samples of each case showed almost the same void structure, and among them, one MIP result is shown in Figure 14 and Figure 15. Under all experimental conditions, the pore distribution shifted to a smaller value due to the strength development according to age regardless of the adjustment of heating temperature, heating time, and cooling time. Particularly, in the range of pore diameters of 0.1–0.01 μm, it was confirmed that the pore distribution of the specimen at the ages of 7 days and 14 days shifted to a smaller value compared to the specimen immediately after the demolding and age of 1 day. Moreover, the number of pores in the specimens at the ages of 7 days and 14 days tended to increase in the vicinity of the capillary pore. The cumulative void volume also confirmed results similar to the void volume results. However, it was confirmed that the porosity of MW3 decreased slightly, although the difference was not significant. The result of the porosity showed that the pore distribution was 16.0% of MW3, 16.2% of MW2, and 16.5% of MW1. The difference in the amount of pores occurred due to the adjustment in the heating temperature and heating time of the MW. Therefore, it was confirmed that the porosity was reduced by adjusting the heating temperature and the heating time of the MWs. As a result, strength enhancement due to the texture densification was confirmed. However, in the case of steam curing in this experiment, the maximum temperature exceeded the set temperature by more than 30 °C, as shown in Figure 9, and the initial temperature control was not performed smoothly. Therefore, it is considered that the change in curing temperature due to the difference in the hydration reaction rate caused the rapid increase in the strength at the early ages rather than the MW heat curing.

Figure 16 shows the result of the SEM observation of hydration products by steam curing and MW3 specimens. In the case of the MW3 specimens, calcium silicate hydrate (C-S-H) was observed in almost all cases with small amounts of Ca(OH)₂. In addition, in the environment where the curing temperature did not exceed 65 °C at the maximum, the acicular-form ettringite hydrate product was barely observed. On the other hand, in the case of steam curing, the temperature control was not smooth, and the initial curing temperature was almost 90 °C, such that a large amount of hydrated products of the acicular form ettringite was observed. However, in the case of steam curing at high temperatures, when the temperature exceeded 70 °C in the hydration process, a problem was observed in the development of long-term strength due to the expansion by the delayed ettringite formation (DEF) phenomenon [15,16]. Therefore, it is necessary to thoroughly control the temperature, and it is considered that the MW heat curing can efficiently control the temperature.

#### 3.2.4. Shrinkage and Cracking Properties

Figure 17 and Figure 18 show the results of free shrinkage and restraint shrinkage, respectively. The amount of deformation was corrected by Equation (1) using the measured strain ($\varepsilon^{\prime }\left(t\right)$), coefficient of linear thermal expansion ($TE_{c}=8.5\;\mu /{}^{\circ}C$), temperature of concrete in time *t* (T(t)), and concrete temperature at the time of casting (T(0)) [17]:(1)$$\varepsilon^{\prime }\left(t\right)=\varepsilon \left(t\right)-TE_{c}\cdot \left(T\left(t\right)-T\left(0\right)\right)$$

The amount of free shrinkage and restraint shrinkage sharply contracted at an early stage as the temperature increased rapidly. It was considered that this occurs from the apparent shrinkage due to the difference in the thermal expansion coefficients of the reinforcing bars attached to both the center and the strain gauge and the strain gauge according to the rapid temperature change [18]. Thereafter, the hydration reaction expanded the concrete. After the demolding, the concrete shrank again, but the overall shrinkage was small. This is because the water–cement ratio of the applied concrete was low, limestone aggregate was used as the coarse aggregate, and the humidity was high due to the execution of the experiment in an indoor environment.

To determine the crack resistance, we calculated the restraint stresses generated in the test specimen from Equation (2) using Young’s modulus ($E_{s}$), restraint deformation ($\varepsilon_{s}$), sectional area of reinforcing bars ($A_{s}$), and sectional area of concrete ($A_{c}$) [19,20,21]:(2)$$\sigma =-\left(A_{s}/A_{c}\right)\cdot E_{s}-A\varepsilon_{s}$$

Figure 19 shows the result of restraint stresses. A compressive stress of approximately 0.03–0.24 MPa was introduced in the steam curing and the MW heat curing at 28 days of age because the expansion stress due to rapid hydration at an early age and the overall shrinkage and restraint degree of the concrete were small.

Moreover, the tensile strength was calculated from the compressive strength ($F_{c}$) of concrete by Equation (3), and the crack generation strength ($f_{cr}$) of concrete was calculated from Equation (4). Further, the crack strength ratio ($R_{cr}$) was calculated from Equation (5) by the strength ratio between the crack generation strength and the restraint stress [20,21].
(3)$$\sigma_{B}\left(t\right)=0.291\cdot F_{c}{\left(t\right)}^{0.658}$$
(4)$$f_{cr}\left(t\right)=0.7\cdot \sigma_{B}\left(t\right)$$
(5)$$R_{cr}\left(t\right)=\sigma_{B}\left(t\right)/f_{cr}\left(t\right)$$

Figure 20 shows the calculation result of the crack strength ratio. Because the crack strength ratio was very low in both steam curing and MW heat curing, the possibility of cracking due to shrinkage was very low in this test condition. In addition, the MW heat curing was considered to have no particular problem in the cracking property due to shrinkage as compared with the steam curing.

#### 3.2.5. Surface Condition Assessment

The surface condition of the concrete after steam curing and MW heat curing was observed by visual inspection. Table 6 shows the surface photograph of concrete at 28 days of age. In the case of steam curing, the surface of the concrete was not smooth due to the influence of moisture generated by the steam and the bleeding of the concrete, and the separation phenomenon was confirmed. This is expected to lead to additional work such as impregnation and reapplication at the time of shipment of PCa products. However, in the case of MW heat curing, no phenomena such as separation were observed at any level, and the surface condition of the concrete, as a whole, was good.

In order to measure the surface defective condition of the concrete by bleeding, the air permeability was measured by the Torrent method [22,23] under the condition that the surface moisture content of the concrete was constant. Figure 21 shows the results of the Torrent air permeability at 28 days. The Torrent air permeability was lower than 1 × 10^−17^ for both the steam curing and MW3, but slightly lower for the MW heat curing. It is considered that the surface condition of concrete is not a problem as compared with that of steam curing.

#### 3.2.6. Environmental and Economic Evaluation

The CO_2_ emissions are simply calculated by Equations (6) and (7) because kerosene was used for steam curing. In the case of the MW heat curing, it was calculated by Equation (8) based on the amount of power. The energy cost of curing was calculated by Equation (9) for steam curing and Equation (10) for MW heat curing.
(6)$${\mathrm{CO}}_{2}\;\mathrm{Emissions}=\mathrm{Fuel}\;\mathrm{consumption}\cdot \mathrm{Unitcalorific}\;\mathrm{value}\cdot \mathrm{Carbon}\;\mathrm{emmision}\;\mathrm{factor}\;\cdot \left(44/12\right)$$
(7)$${\mathrm{CO}}_{2}\;\mathrm{Emissions}\;\mathrm{per}\;1{\;\mathrm{kg}\;\mathrm{of}\;\mathrm{concrete}}_{\left(\mathrm{steam}\;\mathrm{curing}\right)}={\mathrm{CO}}_{2}\mathrm{Emissions}/\mathrm{Total}\;\mathrm{production}$$
(8)$${\mathrm{CO}}_{2}\;\mathrm{Emissions}\;\mathrm{per}\;1{\;\mathrm{kg}\;\mathrm{of}\;\mathrm{conrete}}_{\left(\mathrm{MW}\;\mathrm{heat}\;\mathrm{curing}\right)}=\mathrm{Electricity}\;\mathrm{consumpution}\cdot \mathrm{Emmision}\;\mathrm{factor}/\mathrm{Concrete}\;\mathrm{weigh}$$
(9)$$\mathrm{Curing}\;\mathrm{energy}\;\mathrm{cost}\;\mathrm{per}\;1{\;\mathrm{kg}\;\mathrm{of}\;\mathrm{conrete}}_{\left(\mathrm{steam}\;\mathrm{curing}\right)}=\mathrm{Purchase}\;\mathrm{price}\;\mathrm{of}\;\mathrm{heavy}\;\mathrm{oil}\cdot \mathrm{Fuel}\;\mathrm{consumpution}\cdot \mathrm{Total}\;\mathrm{production}$$
(10)$$\mathrm{Curing}\;\mathrm{energy}\;\mathrm{cost}\;\mathrm{per}\;1{\;\mathrm{kg}\;\mathrm{of}\;\mathrm{conrete}}_{\left(\mathrm{MW}\;\mathrm{heat}\;\mathrm{curing}\right)}=\mathrm{Electricity}\;\mathrm{consumpution}\cdot \mathrm{Electricity}\;\mathrm{fee}\cdot \mathrm{Concrete}\;\mathrm{weigh}$$

The unit calorific value of the heavy oil and the carbon emission factor are the values provided by the Japanese Ministry of the Environment, the fuel consumption and gross weight are the values provided by A Company, the emission factor is the Hokkaido power value, and the electricity usage and concrete weight were calculated based on the values used in this experiment.

Figure 22 shows the CO_2_ emission and curing energy cost results when applying steam curing and MW heat curing. The CO_2_ emissions were estimated to be reduced by approximately 51% for MW1, approximately 36% for MW2, and approximately 52% for MW3 compared to that using steam curing. The energy cost of curing 1 kg of concrete is expected to be reduced by approximately 28% in MW1, approximately 7% in MW2, and approximately 29% in MW3 compared to that using steam curing.

## 4. Conclusions

In this study, the performance of precast concrete using a microwave heating system was evaluated, and the following conclusions were obtained.
MW heat curing can efficiently control the temperature of concrete within a range of ± 5 °C, and it can satisfy the demolding strength of concrete after curing by densification of the concrete matrix by decreasing the porosity.It is necessary to thoroughly control temperature to prevent DEF under a high-temperature environment, and it is considered that MW heat curing can efficiently control the temperature.When MW heat curing is applied, superior performance—such as cracking due to shrinkage, the prevention of bleeding of concrete after curing, and the prevention of surface separation—can be expected, unlike that with steam curing.MW heat curing can be expected to reduce CO_2_ emissions and curing energy cost compared with steam curing.To apply MW heat curing more efficiently as a substitute for steam curing, it is necessary to verify and supplement with actual data about the various sizes and shapes of steel form.

## Figures and Tables

**Figure 1 materials-12-01113-f001:**
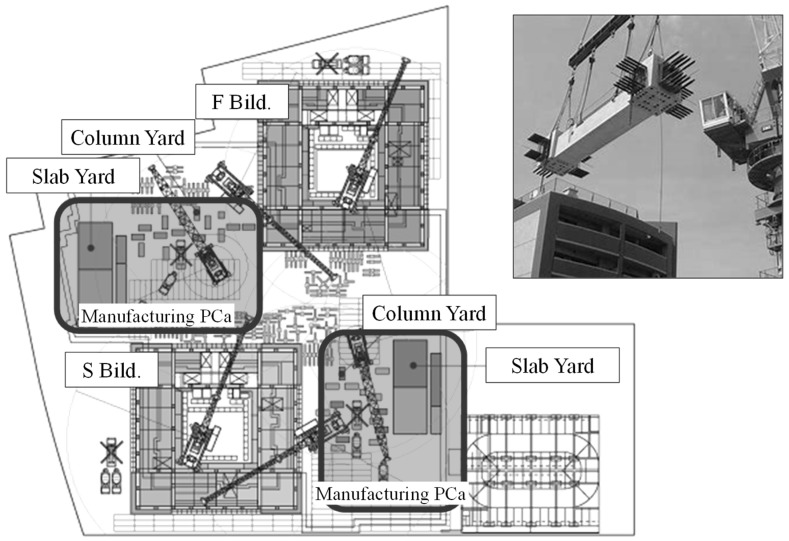
Schematic of site precast concrete (PCa) method.

**Figure 2 materials-12-01113-f002:**
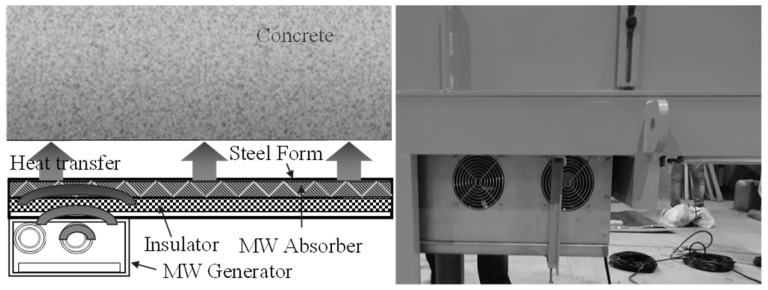
Heating of a steel form by microwaves (MWs).

**Figure 3 materials-12-01113-f003:**
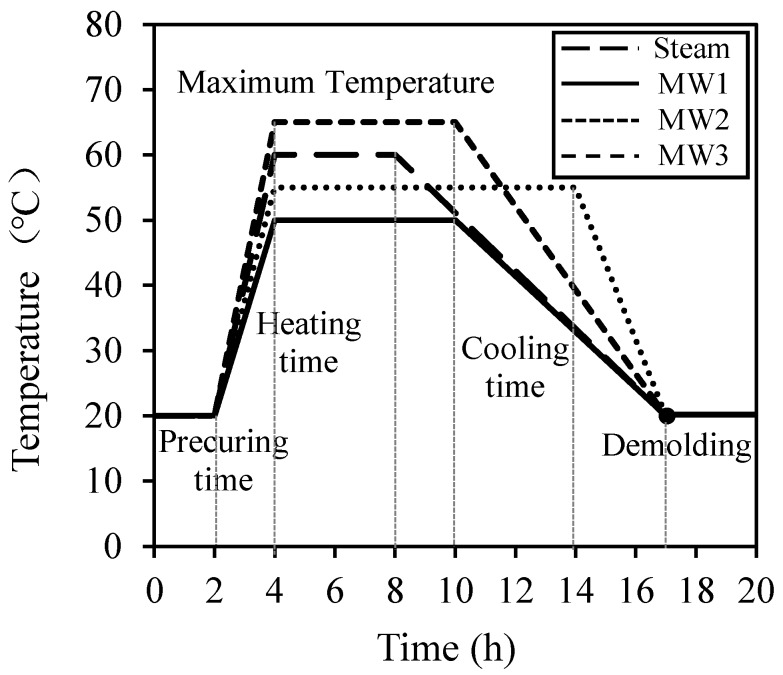
Curing cycle.

**Figure 4 materials-12-01113-f004:**
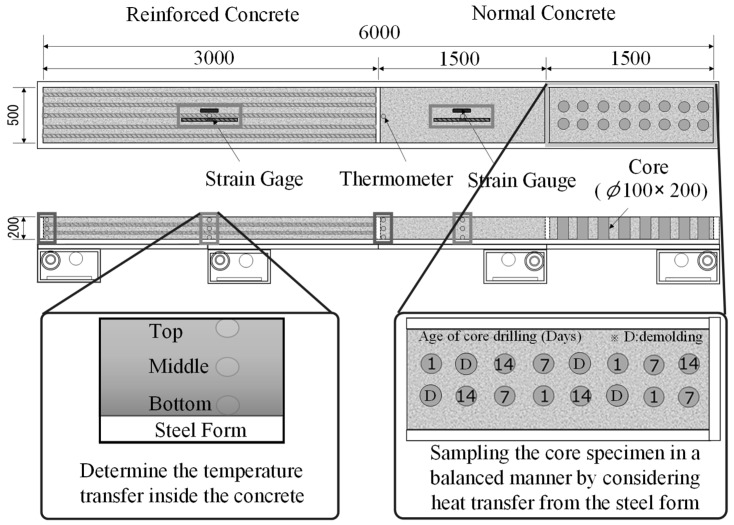
Outline of mock-up specimen.

**Figure 5 materials-12-01113-f005:**
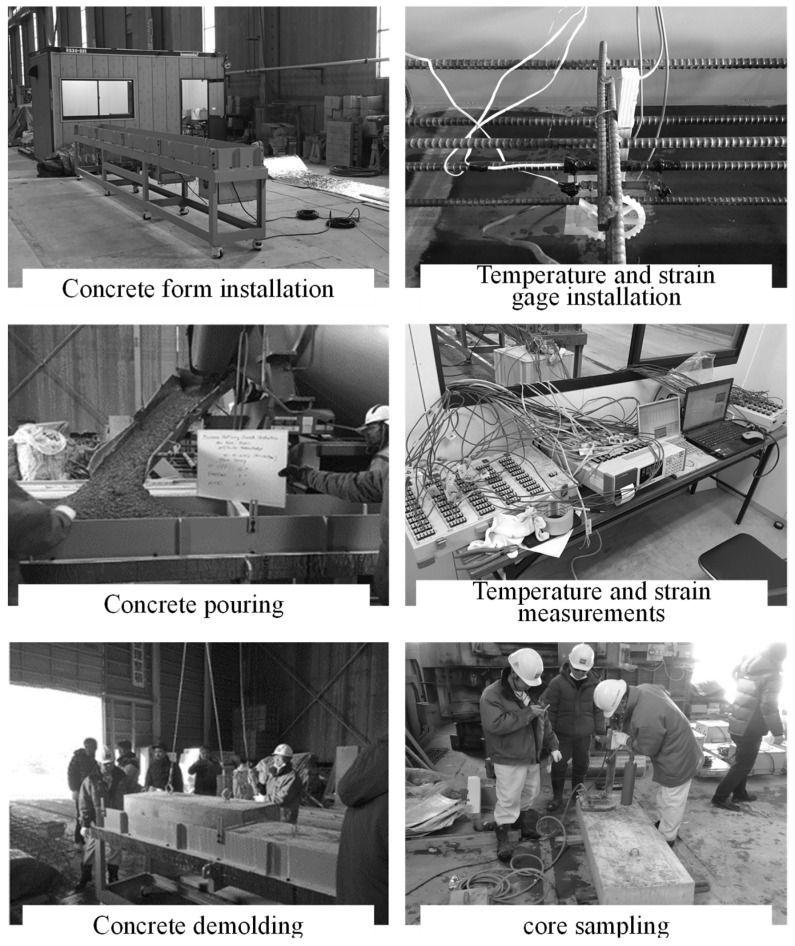
Experimental setup of mock-up test.

**Figure 6 materials-12-01113-f006:**
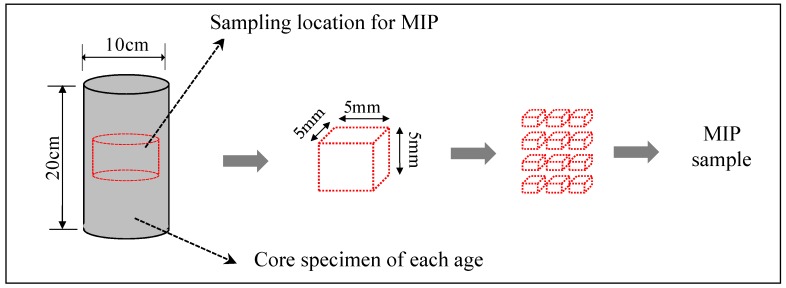
Sampling of a specimen for mercury intrusion porosimeter (MIP).

**Figure 7 materials-12-01113-f007:**
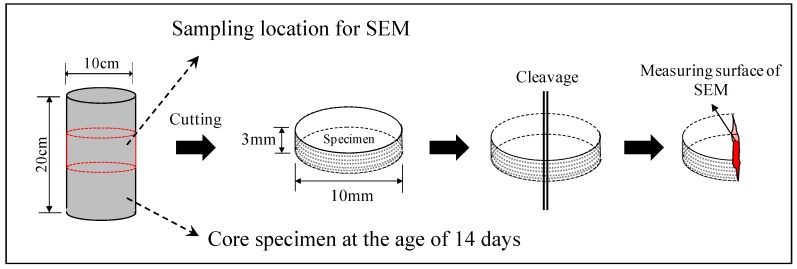
Sampling of a specimen for scanning electron microscopy (SEM).

**Figure 8 materials-12-01113-f008:**
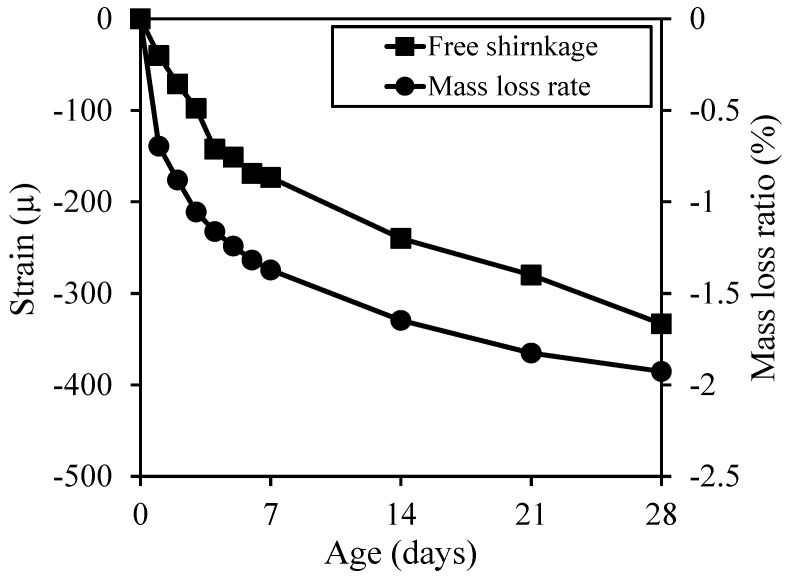
Free shrinkage and mass loss ratio.

**Figure 9 materials-12-01113-f009:**
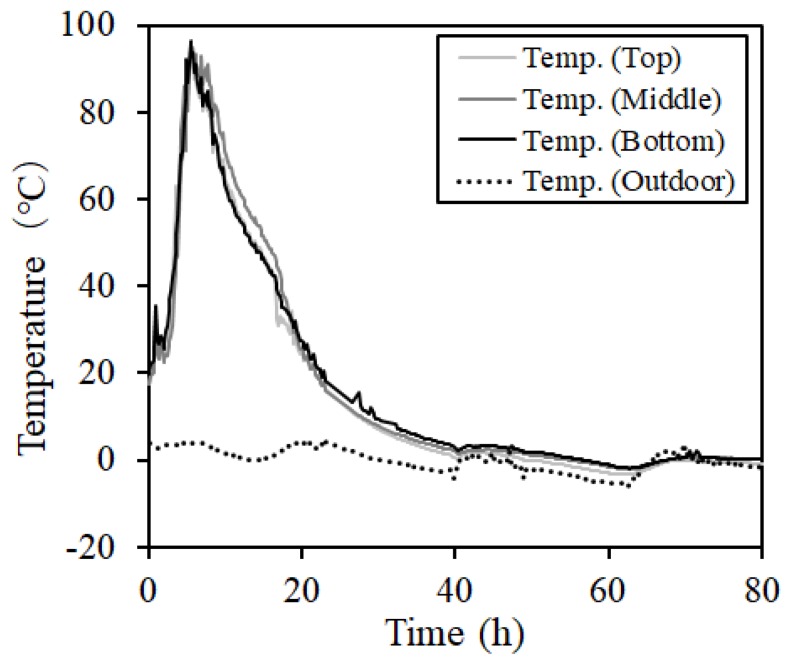
Concrete temperature distribution (steam curing).

**Figure 10 materials-12-01113-f010:**
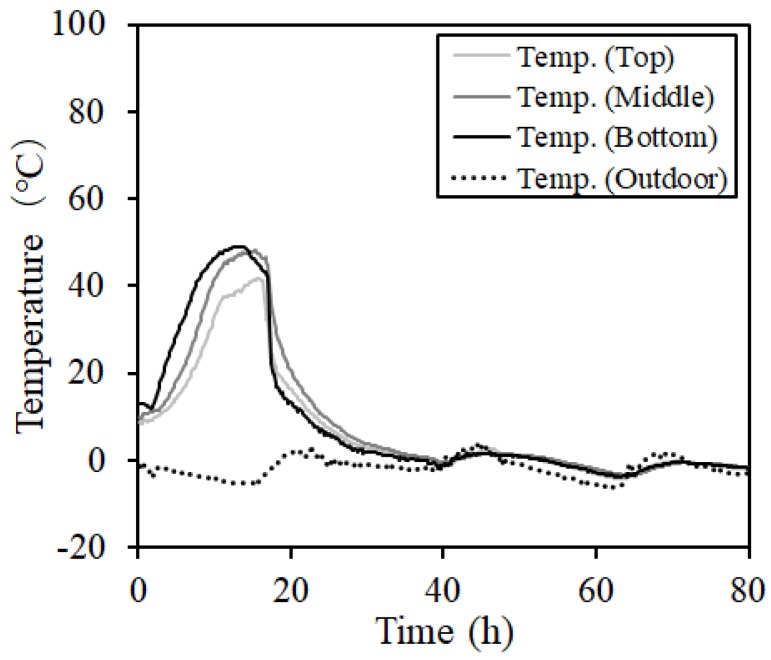
Concrete temperature distribution (ex: MW2).

**Figure 11 materials-12-01113-f011:**
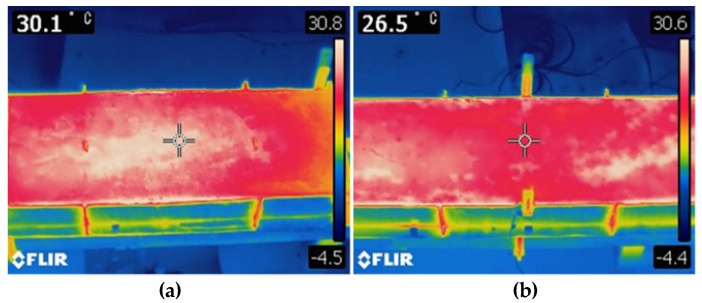
Concrete surface temperature distribution. (**a**) Steam curing (**b**) MW heat curing (ex: MW2).

**Figure 12 materials-12-01113-f012:**
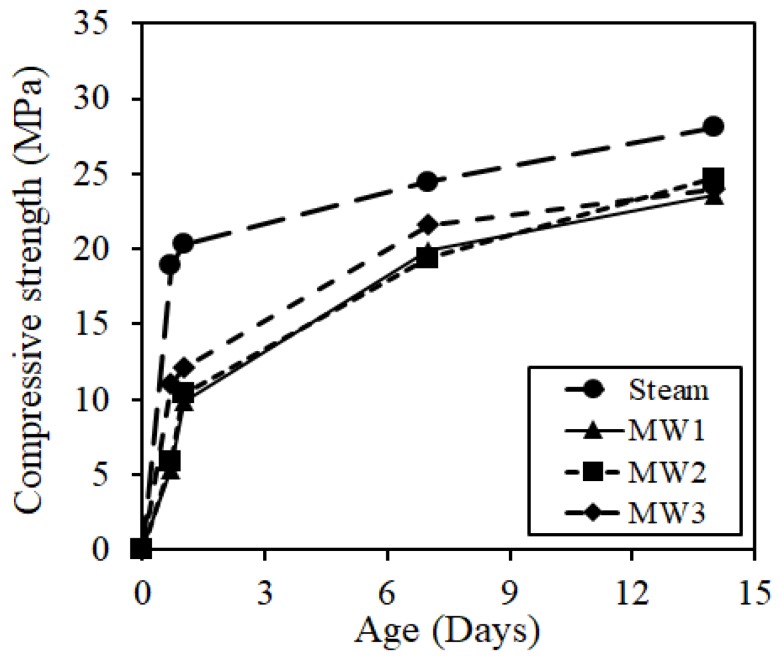
Compressive core strength.

**Figure 13 materials-12-01113-f013:**
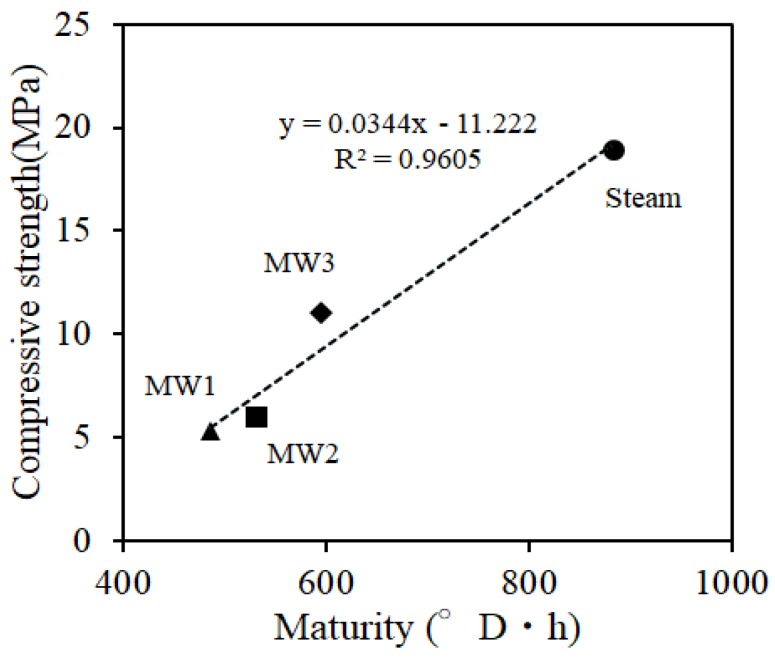
Relation between compressive core strength and maturity.

**Figure 14 materials-12-01113-f014:**
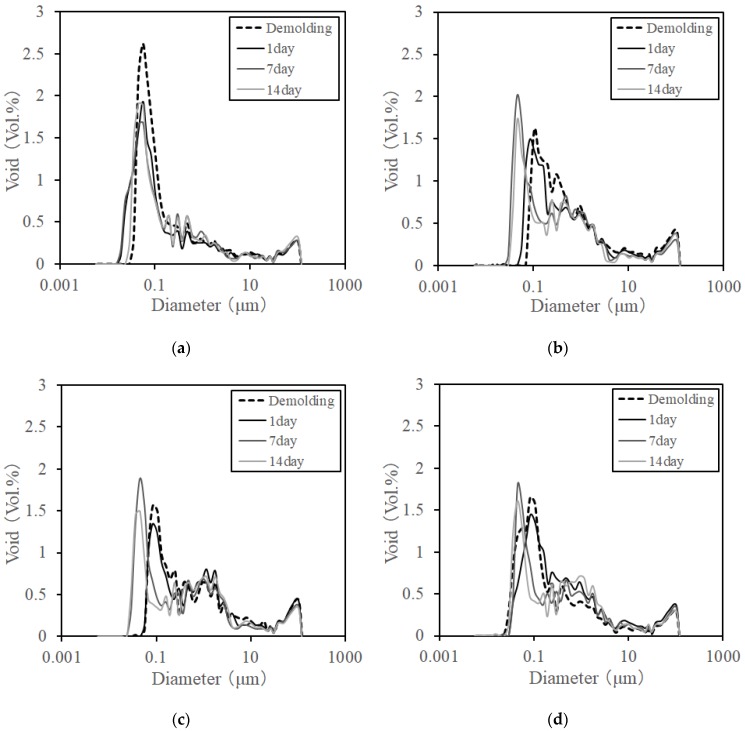
Porosity of concrete. (**a**) Steam curing, (**b**) MW1, (**c**) MW2, (**d**) MW3.

**Figure 15 materials-12-01113-f015:**
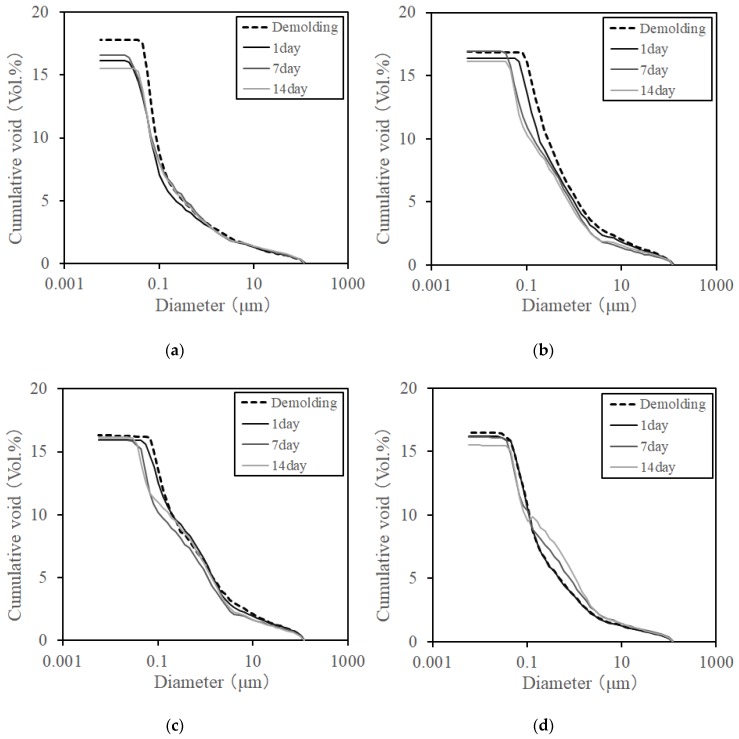
Cumulative porosity of concrete. (**a**) Steam curing, (**b**) MW1, (**c**) MW2, (**d**) MW3.

**Figure 16 materials-12-01113-f016:**
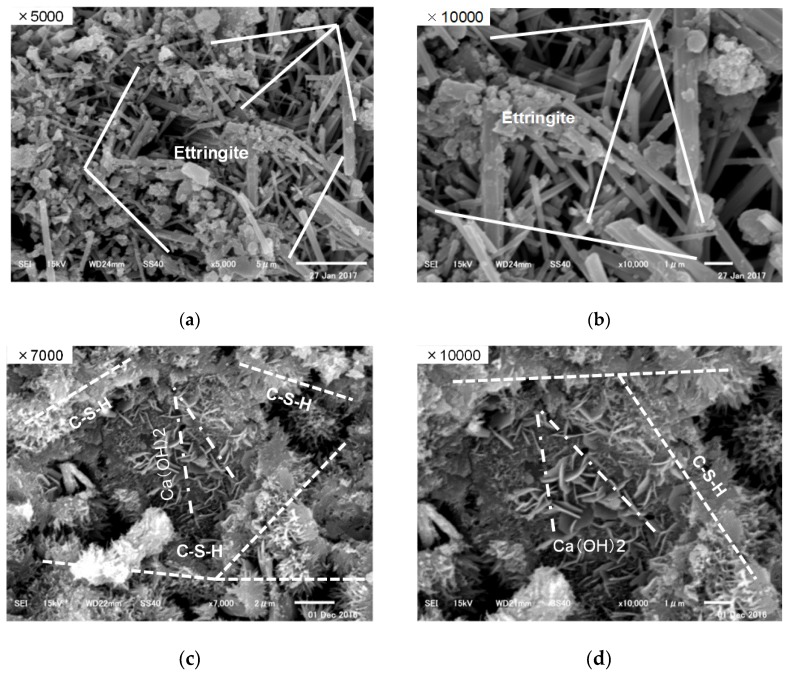
SEM observation. (**a**) Steam curing (×5000), (**b**) Steam curing (×10,000), (**c**) MW 3 (×7000), (**d**) MW 3 (×10,000).

**Figure 17 materials-12-01113-f017:**
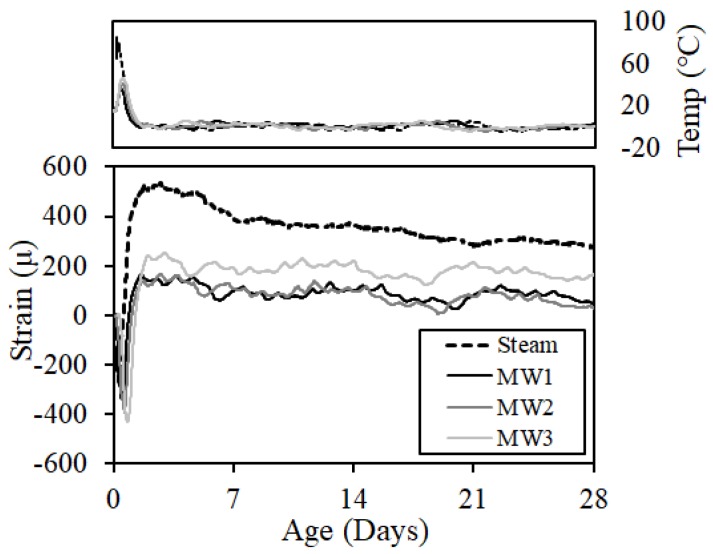
Free shrinkage.

**Figure 18 materials-12-01113-f018:**
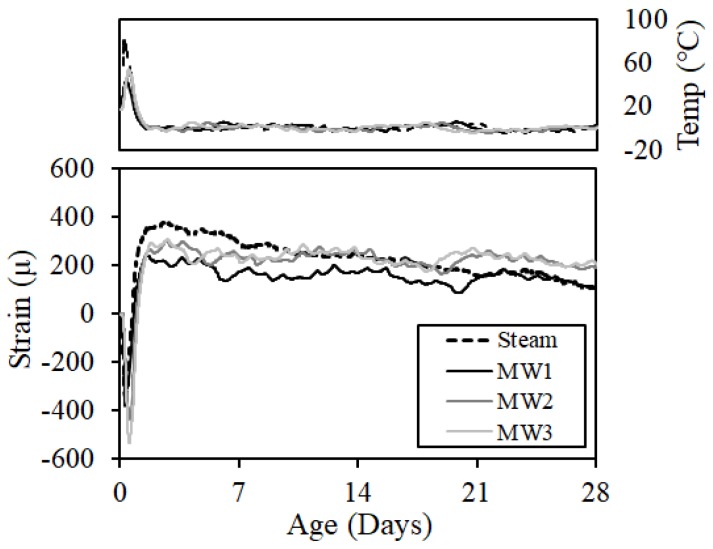
Restraint shrinkage.

**Figure 19 materials-12-01113-f019:**
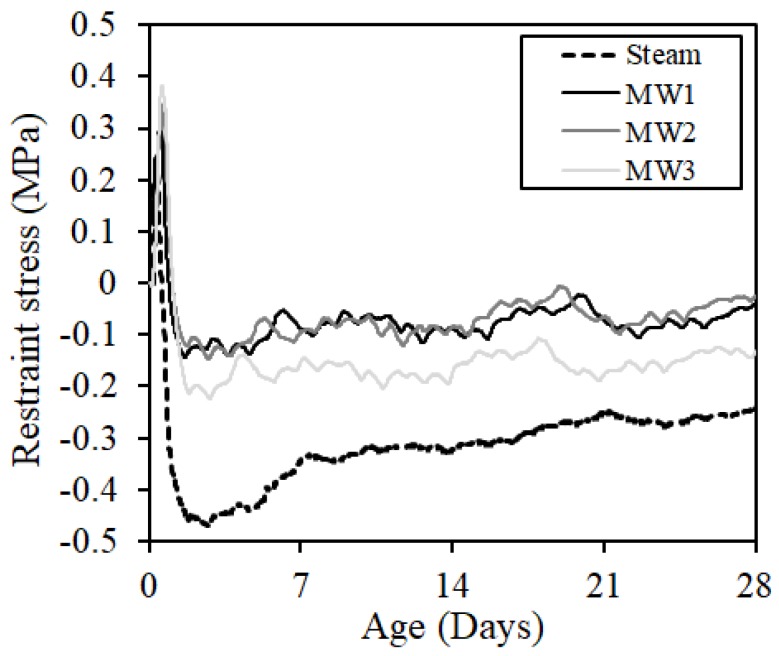
Restraint stress.

**Figure 20 materials-12-01113-f020:**
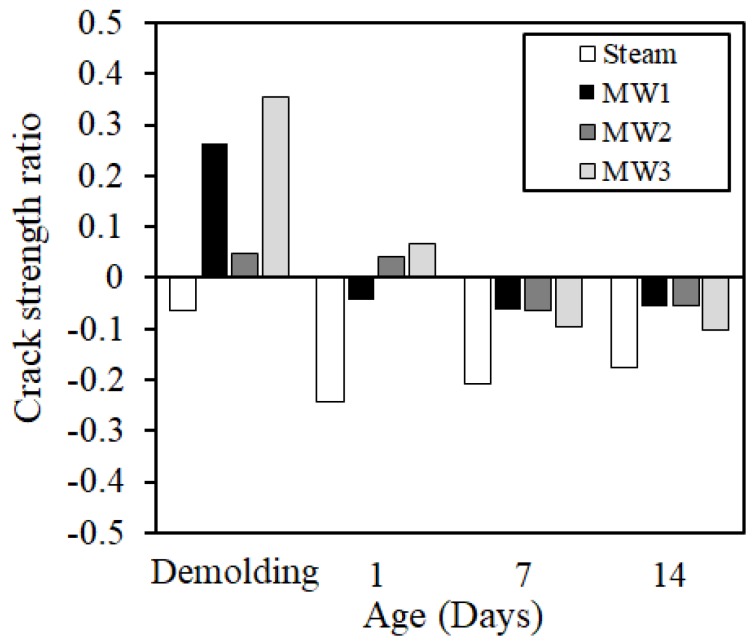
Crack strength ratio.

**Figure 21 materials-12-01113-f021:**
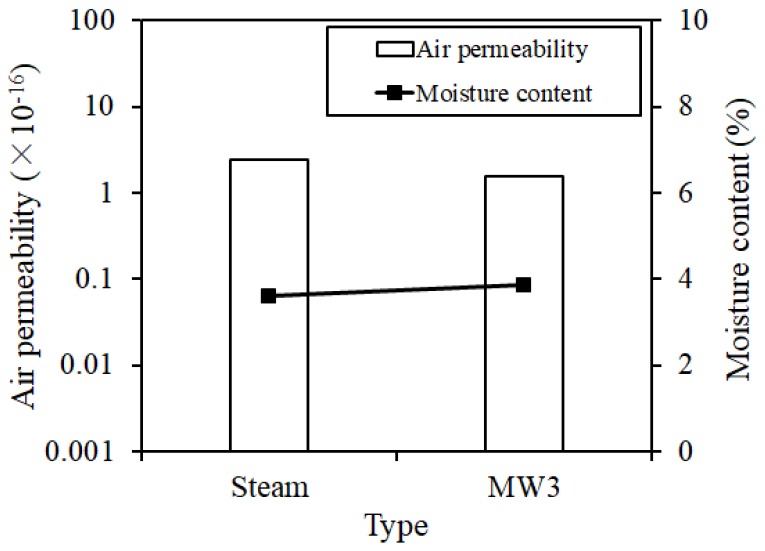
Torrent air permeability.

**Figure 22 materials-12-01113-f022:**
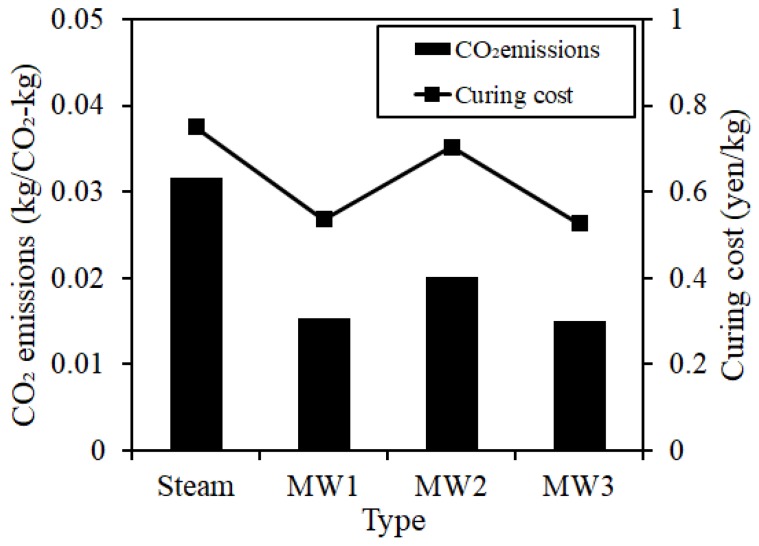
CO₂ emission and curing energy cost.

**Table 1 materials-12-01113-t001:** Experimental conditions.

Type	W/C ^1^ (%)	Slump (cm)	Air (%)	Curing Time ^2^ (h)	Maximum Temperature (°C)	Evaluation Items
Steam	42.6	18.0 ± 2.5	4.5 ± 1.5	2, 6, 9	60	Fresh properties (slump, air contents) Temperature distribution Compressive strength Mercury intrusion porosimeter Scanning electron microscopy Free shrinkage Restraint shrinkage Surface performance Air Permeability CO_2_ emission Curing energy cost
MW1				2, 8, 7	50	
MW2				2, 12, 3	55	
MW3				2, 8, 7	65	

^1^ W/C, water-to-cement ratio; ^2^ precuring time, heating time, cooling time.

**Table 2 materials-12-01113-t002:** Materials used.

Materials	Type
Cement	Ordinary Portland cement, Density: 3.16 g/cm^3^
Sand	Pit sand, Density: 2.67 g/cm^3^, Absorption ratio: 1.44%
Gravel	Limestone, Density: 2.71 g/cm^3^, Absorption ratio: 1.05%
Admixture	Air entraining and water reducing agent and Air entraining agent

**Table 3 materials-12-01113-t003:** Chemical composition of cement.

Ordinary Portland cement	**Chemical Composition (%)**								
	**SiO_2_**	**Al_2_O_3_**	**Fe_2_O_3_**	**CaO**	**MgO**	**SO_3_**	**CaSO_4_**	**Ig.loss**	**Alkali content**
	21.4	5.5	2.8	64.3	2.1	1.9	-	0.56	0.25

**Table 4 materials-12-01113-t004:** Concrete mix parameters.

*f_ck_* (MPa)	W/C (%)	s/a (%)	Unit Weight (kg/m^3^)				AE Reducing Agent (kg/m^3^)	AE Agent (kg/m^3^)
			W	C	G	S		
30	42.6	43.6	165	387	819	972	3.87	0.155

**Table 5 materials-12-01113-t005:** Fresh properties and compressive strength.

Curing	Fresh Properties			Compressive Strength			
	Slump (cm)	Air Content (%)	Temperature (°C)	1 Day (MPa)	7 Days (MPa)	14 Days (MPa)	28 Days (MPa)
Steam	18.0	4.8	16.0	7.3	33.2	37.9	40.8
MW1	18.5	5.3	13.0				
MW2	19.0	5.3	11.2				
MW3	19.0	4.7	15.2				

**Table 6 materials-12-01113-t006:** Surface condition of concrete at the age of 28 days.

**Type**	**Steam**	**MW1**
Shape of surface	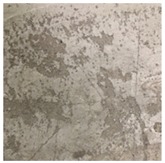	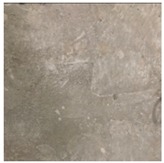
**Type**	**MW2**	**MW3**
Shape of surface	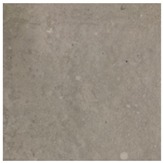	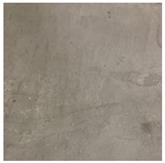

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
