# Peer review of "Performance Evaluation of Precast Concrete Using Microwave Heating Form"

_materials, 2019, doi:10.3390/ma12071113_

Round 1
Reviewer 1 Report
The review considers the manuscript entitled: „Performance Evaluation of Precast Concrete Using Microwave Heating Form”. The manuscript refers to the evaluation of basic physical and geometrical parameters of precast concrete element cured using microwave. The authors states that the following properties are investigated: temperature distribution, strength development, porosity, microstructure using SEM, shrinkage and surface properties. The details of precast site and heating form are also presented. The manuscript is interesting but major improvements have to be introduced before further evaluation of the manuscript. The mistakes or issues, which have to be clarified are listed below:
1. Give information about size of limestone (grading curve),
2. To analyze the shrinkage of concrete the detailed information on cement chemical composition is needed,
3. Line 132 there is no unit of shrinkage,
4. Figure 9 gives no important information as the kind of concrete is not specified,
5. The reviewer suppose that mercury intrusion porosimetry (MIP) analysis is incorrect. For any test to be correct the result representative requirements must be met. Therefore the samples volume is limited. For MIP test we usually apply the sample of 1-5 cm3. The representative volume for concrete , depending on the aggregate size, is usually more than 1 dm3. Therefore the results obtained by the authors are not representative for their concretes. Additionally the authors didn’t specify the apparatus. Concluding the section devoted to MIP analysis must be canceled.
6. The results presented in Figure 14 (SEM analysis gives only local results) do not proof that the amount of ettringite in Steam curing concrete is larger than in MW. This must be literally stated in the manuscript.
Author Response
We would like to thank the reviewer for carefully reading and for giving quite valuable comments and suggestions, which substantially helped improving the quality of this manuscript. We describe our response point by point to each comment (in bold letters) and we have marked the modifications in red in the paper. Please refer to attachment file.

Reviewer 2 Report
The authors describe the possibility of microwave curing to control the temperature (and resultant properties) of precast concrete. The research is interesting and novel but the paper needs some improvement. There is a lack of information of some figures and some tables are a bit strange.
Many figures (like Fig 1 and 5) cannot be understood without extra information. Please add this information. It can be done in the text, but it would be better to add info to the legend of the figure. If we want to improve the quality of the journal, we should make every figure self standing. That means that with just the figure and legend, it can be understood.
Table 1-3: these are strange tables. Only part of the info needs to be put in a table. The rest can (or is already) in the text.
Fig 3 It is not clear which temperature is given here. Even after reading the paper I still don’t know. For MW I think it is the preset temperature, but for the steam curing, it deviates a lot from the measured temperature.
Table 4 why no compressive strength data for steam to MW2 samples?
Fig 6 the rate is the derivative! this is not the rate.
Fig7 you did these experiments at an outside temperature of about 0°C???
L129 >30MPa at 7 days and only 10 at 28days??? Furthermore, which sample are you talking about? Only for MW3 data are given.
L144 In the figure the temperature goes to 90°C, that is 30°C higher than 60°C!
L178 How many samples were used for the pore size distribution? How repeatable are these measurements?
L195 Strange sentence. The void content will not cause the rapid increase in strength (but the cure temperature does)
L269 here kerosene, in equation 13 it is heavy oil?
L277 in this cost you only take the working cost, not the cost of the installation. How does those relate? Are the costs for microwaves and steam curing comparable?
Conclusions: your conclusion is that microwave curing has a lower CO2 emission, but in fact this does not depend only on microwave or steam curing, but on the way the energy is produced. For steam curing also another heat source could be used (like solar power) which can make it less environmental polluting. I would rather like to see a conclusion based on the energy that is used. What if a combination of heat and electricity generation is used? In that way you also reduce the environmental impact. So please make your conclusion more general.
See pdf for some extra comments:
Table 1-3: For table 1 only curing time and maximum temp have to be in the table.
L63 and Fig 3 there is no information on how the temperature is controlled (or is there no control?).
L104 ‘The compressive strength and porosity of the specimens were determined by a mercury intrusion porosimeter (MIP).’ Please rephrase as compressive strength is not measured by MIP.
L129 ‘The compressive strength exceeds the design strength of 30 MPa at 7 days of age. In addition, it exceeds the design strength of 10 MPa at 28 days of age.’ More than 30MPa at 7 days and only 10 at 28days??? This sentence needs to be rephrased.
L160 ‘Fig. 10 shows the compressive strength of the core specimen, which is a simulated member.’ what do you mean by ‘simulated member’?
L257-258: not clear what you mean. The MW are then MW1 and 2?
Eq. 12 and 13: Weight instead of Weigh

Author Response

(The authors gave the same response as above.)

Reviewer 3 Report
This paper presents results from a novel method where microwave heating is applied to cure concrete. This work has merit, nicely written and can be suitable for publications once the following suggestions are incorporated.
1) Overall, please provide a general discussion on how MW differ than steam curing from the point of view of material properties. Will MW improve thermal and mechanical properties of concrete? How about durability and environmental resilience?
2) Some of the properties in Table 4 are missing. Please provide these values to enable a comprehensive comparison between the different suing methods used herein.
3) Please discuss the amount of energy required to activate and sustain the microwave energy. How does this energy cost compare to that of conventional steaming process currently adopted by the precast industry?
4) On a similar note, it seems that there will be a need to design and develop specialized industrial-rated microwaves for use in sites. Is this true? Or perhaps portable microwaves can be used?
5) What are the requirements for such microwaves in terms of heating capacity, heating uniformity, etc.?
6) Figure 10 shows that steam curing exhibits higher initial strength compared to MW heat curing by about 10%. In some cases, 10% might delay construction >12 hours. How to improve the effectiveness of MW heating? Is it possible/feasible?
7) Does the CO2 calculation include the residue effects needed from using electricity to operate the MW?
8) As a side noted, curing through MW, rather than water/steam, seems to be very attractive for space construction applications as noted by past and recent researchers. This might help open up a new opportunity to the authors in expanding their proposed technique.
a. https://doi.org/10.1061/40177(207)82
b. https://doi.org/10.1016/j.paerosci.2018.03.004
Author Response

(The authors gave the same response as above.)

Round 2
Reviewer 1 Report
I would like to thank the authors for the amendments, which considerably improved the quality of the manuscript. However, according to my opinion, one think remains incorrect. Namely, the authors state that they use “Fine aggregate and coarse aggregate (maximum dimension: 20 mm) were used for crushed sand and limestone crushed stone”. Then describing the process of sample preparation they give the dimension of sample for MIP test as “samples were collected for each age and cut into cubes of 5 mm”.
Concluding the dimension of sample made of concrete for MIP test is much smaller than the maximum dimension of aggregates. This numbers completely violates the regime of reliable experiment performance.
Generally speaking we usually assume that the representative diameter of cubic sample made of concrete should be around 5 times larger than the maximum aggregate. The test performed using such sample give the representative results. The presented MIP test doesn’t meet those requirements therefore I cannot accept its results.
Reviewer 3 Report
Thank you for your efforts.